# Long-COVID symptom monitoring: Insights from a two-year telemedicine study

**Andrea Foppiani[1,2], Chiara Montanari[3,4], Sara Zanelli [4], Michele Davide Maria Lombardo[5], Valeria Calcaterra [4,6] *, Gianvincenzo Zuccotti[3,4]**

**1** Department of Food, Environmental and Nutritional Sciences (DeFENS), International Center for the Assessment of Nutritional Status (ICANS), University of Milan, Milan, Italy, **2** Department of Endocrine and Metabolic Medicine, Clinical Nutrition Unit, IRCCS Istituto Auxologico Italiano, Milan, Italy, **3** Department of Biomedical and Clinical Science, University of Milan, Milan, Italy, **4** Pediatric Department, "V. Buzzi" Children's Hospital, Milan, Italy, **5** Orthopaedics and Traumatology, University of Milan, Milan, Italy, **6** Department of Internal Medicine, University of Pavia, Pavia, Italy

* valeria.calcaterra@unipv.it

**Data Availability Statement:** All relevant data are within the manuscript.

**Funding:** The project received contributions from (1) Bando Cariplo Networking research & training

## Abstract

### Background

The diverse manifestations of Long-COVID have become increasingly important due to their significant impact on patients' lives. Telemedicine has emerged as an important tool for post COVID-19 follow-up. This study is part of a large cohort study involving COVID-positive patients monitored by the COD19 telemedicine platform operations center. We recontacted patients who were initially monitored from February 2020 to May 2020 to assess the presence of Long-COVID symptoms at a 2-year follow-up.

### Methods

We conducted interviews to evaluate Long-COVID symptoms at the 2-year mark and investigated whether patients had contracted a second COVID-19 infection between the 1-year and 2-year follow-ups, and recorded their vaccination status.

### Results

Out of 165 patients, 139 (84%) reported symptoms at the 1-year follow-up, while only 101 (61%) reported symptoms at the 2-year follow-up. Among patients with Long-COVID symptoms at the 2-year follow-up, the majority (80, 49%) had experienced Long-COVID at the 1-year follow-up, received the SARS-CoV-2 vaccine, and had not experienced a second infection between the two follow-ups. Both having Long-COVID at the 1-year follow-up and contracting a second infection were significant risk factors for presenting with Long-COVID at the 2-year follow-up.

### Conclusions

To the best of our knowledge, this study stands out as one of the few that includes a 2-year follow-up on Long-COVID symptoms using telemedicine. Telemedicine has proven to be an

post-COVID protocol number 2021–4490; (2) HORIZON-HLTH-2021-CORONA-01 CoVICIS project number 101046041; (3) PRIN: research projects of significant national interest - 2022 call prot. 20228pnnjl.

**Competing interests:** The authors have declared that no competing interests exist.

effective and innovative tool for long-term patient monitoring, early diagnosis, and treatment. Telemedicine represents a significant future challenge for healthcare.

## Introduction

Since 2020, COVID-19 has affected millions of people around the world, causing not only acute illness but also a condition known as Long-COVID characterized by persistent symptoms. Long-COVID is an emerging disorder that is often underestimated and undertreated, yet its varied manifestations significantly impact patients' lives.

The World Health Organization (WHO) defines Long-COVID as the persistence or emergence of new symptoms 3 months post-acute infection, lasting for at least 2 months without any alternative explanation [1]. In 2022 the National Institute for Health and Care Excellence (NICE) further refined this definition, identifying two conditions: ongoing symptomatic COVID-19, lasting from 4 to 12 weeks, and post COVID-19 syndrome, lasting 12 weeks or more [2].

Long-COVID is estimated to affect at least 10% of those infected with SARS-CoV-2 [3]. Its occurrence is not dependent on the severity of the initial infection; even asymptomatic or mildly affected individuals can develop Long-COVID, though their risk is lower. The pathophysiology of Long-COVID remains unclear, with hypotheses including persistent viral presence, autoimmunity, microbiome dysbiosis, and tissue damage [4]. More research is needed to fully understand and treat this condition.

Long-COVID is a complex multisystemic condition affecting cardiovascular, respiratory, gastrointestinal, endocrinological, neuropsychiatric, musculoskeletal, dermatological, and genitourinary systems. Common symptoms in adults include fatigue, dyspnea, cognitive disorders (e.g., brain fog), myalgia, joint pain, and depression. This symptom diversity may be due to the widespread presence of the angiotensin converting enzyme 2 (ACE2) receptor, the entry point for SARS-CoV-2, in the epithelium of many organs [5]. It is important to highlight that there are no specific treatments for this syndrome up to date. Some interventions seem to have a positive impact in particular, physical activity with targeted exercises, nutritional support, symptomatic medications (e.g., non-steroidal anti-inflammatory drugs) and psychological support. Vaccination could indirectly reduce the probability of developing Long-COVID reducing the risk of SARS-CoV-2 infection [6].

The significant impact of Long-COVID on quality of life and daily functioning necessitates monitoring patients post-acute infection to detect and manage symptoms early through a multidisciplinary approach.

Telemedicine has proven to be a crucial tool for post COVID-19 follow-up. During the pandemic, we introduced a telemedicine platform for an active patient home-care surveillance. Initially this was a project by the University of Milan's Faculty of Medicine, in collaboration with ATS of Milan and ASST Fatebenefratelli-Sacco, designed to provide active home surveillance during the COVID-19 emergency. Named COD19 ("discharged operation center"), this virtual "ad hoc" platform, aimed to monitor and manage clinical symptoms, verify isolation, and address the needs of discharged patients as well as healthcare workers with COVID-19. The service was expanded to cover Milan's major public hospitals and, during the second wave, COD19 enabled monitoring of COVID19 positive patients who were not hospitalized, focusing on those at higher risk of complications. The success of COD19 demonstrated telemedicine's potential and led to the evolution of a full virtual hospital called "COD20—Home

Hospital Care", offering patient-specialist video consultations [7]. COD20 represents a virtual hospital model designed to ensure continuity of care at home through technological solutions, later used to monitor disease progression and manage patient care.

Telemedicine is an integrative healthcare application modality that complements traditional medicine. It enables remote healthcare delivery, aiming to reduce hospital admissions or enable early discharge through close monitoring of the patient's health status [8]. This approach is particularly useful in acute diseases, such as COVID-19, where patients may require ongoing surveillance during acute infection or post-discharge to detect warning symptoms and signs and reduce the risk of complications or flare-ups.

Telemedicine can be further through telemonitoring, which involves the use of sensors (e.g., oxygen saturation), to collect data remotely [9].

Additionally, telemedicine is valuable in diagnostic pathways and during follow-up of chronic patients, improving care quality and assistance [10].

According to the recent guidelines (2020) by the Italian Ministry of Health, telemedicine is now considered an integral part of the National Health Service, contributing to cost optimization and having a significant psycho-social impact [8].

This study is a continuation of a larger cohort study involving COVID-positive patients monitored through the COD19 telemedicine platform. We followed up with patients who were initially monitored from February 2020 to May 2020 and later recontacted them to investigate the presence of Long-COVID symptoms 2-years later. The objective of this report is to share our experience and highlight the potential value of telemedicine platforms in epidemic situations and other conditions where isolation is necessary.

## Methods

### Study design and patient's selection

We initially included all patients with a confirmed COVID-19 diagnosis, verified by a positive molecular swab and polymerase chain reaction positive for viral ribonucleic acid. All patients were discharged by the monitoring service after reaching clinical stability and obtaining 2 negative SARS-CoV-2 swabs within 24 hours of each other. All patients that were included in the 1-year follow-up study were also recontacted for this 2-year follow-up study.

Exclusion criteria were as follows: patients who died during the follow-up or after discharge; patients hospitalized at the time of follow-up; patients under the age of 18; patients with psychiatric disorders; patients who refused to participate in the study; and patients who could not be contacted.

Eligible patients were consecutively contacted, and, after expressing consent to participate in the study, a semi structured interview was administered by phone by trained medical staff from the COD19 operations center between February and March 2022.

The study adhered to the guidelines of the Declaration of Helsinki and was approved by the Ethics Commission of the University of Milan, (Ethics Commission number: 126/20). Written informed consent was obtained from all participants.

### Semi structured interview

The 2-year follow-up interview was based on the results from our previous study, focusing on the most frequently reported symptoms in the literature. A total of 37 items were surveyed and grouped into the following categories: respiratory disorders, fatigue and weakness, muscle and joint pain, movement impairments, neurological and cognitive impairments, sensory alterations, sleep disorders, and gastrointestinal symptoms.

We also explored whether the patient contracted a second COVID-19 infection between the 1-year and the 2-year follow-up, and their vaccination status.

All data regarding the acute phase of the disease were collected during monitoring by the COD19 operation center and either retrieved from electronic clinical records or by directly interviewing the patient when information was unavailable. Data included date of birth, sex, body mass index (BMI), smoking habits, symptoms at onset, admission to the hospital, type of hospitalization (in ward or in intensive care unit [ICU]), medications taken during hospitalization and after hospital discharge (corticosteroids, antivirals, antibiotics, anticoagulants), and comorbidities (hypertension, diabetes, cardiovascular diseases, malignant tumors, chronic obstructive pulmonary disease, and chronic kidney disease).

Disease severity during the acute phase, and any subsequent infection was graded on a 7-level scale according to Huang et al. [11]: (1) discharged from the emergency room, asymptomatic or with mild symptoms; (2) discharged from the emergency room, with symptoms; (3) hospitalized, not requiring supplemental oxygen; (4) hospitalized, requiring supplemental oxygen; (5) hospitalized, requiring high-flow nasal cannula, noninvasive mechanical ventilation (NIV), or both; (6) admitted to hospital requiring extracorporeal membrane oxygenation, invasive mechanical ventilation, or both; (7) death. BMI categories were defined as follows: underweight, BMI < 18.5 kg/m$^2$; normal weight, BMI 18.5–24.9 kg/m$^2$; overweight, BMI 25.0–29.9 kg/m$^2$; obese, BMI ≥30 kg/m$^2$.

## Statistical analysis

Continuous variables are reported as median and interquartile range (IQR) (25th, 75th percentile), whereas categorical variables are reported as count (fraction).

Participants were categorized into 2 groups according to the Huang et al. scale [10]: 1–2 categories (not hospitalized with or without symptoms) and 3–6 categories (hospitalized, requiring or not oxygen or admitted to ICU).

Multivariable logistic generalized additive models were used to describe the relationship between symptoms at 2-year follow-up (dichotomous, present/not present) and age (continuous), sex (dichotomous, female/male), acute phase severity (dichotomous, scale 1–2/scale 3–6), symptoms at 1-year follow-up (dichotomous, present/not present), second infection (dichotomous, infected/not infected), and vaccination status (dichotomous, vaccinated/not vaccinated). Linearity between outcomes and age was not assumed by using a smooth term with a thin replate regression spline. Statistical analyses were performed in R 4.3.3.

## Results

Out of the 303 patients originally included in the 1-year follow-up study, 165 were successfully contacted and included in the 2-year follow-up study conducted between February and March 2022. The characteristics of these patients are detailed in Table 1.

Sexes were almost equally distributed (53% females, 47% males), and median age was 53 (IQR, 42–62) years. Most patients had a normal weight, with a median BMI of 24.9 Kg/m$^2$ (IQR, 22.6–28.0). Although most patients never smoked (59%), the most prevalent comorbidities were hypertension (28%), cardiovascular diseases (9.1%), and diabetes (7.9%).

At the 2-year follow-up, almost all patients (97%) were vaccinated, but only 80% received 3 doses of the vaccine.

Only 10% of patients contracted a second SARS-CoV-2 infection, and their severity was generally low, never exceeding scale 2 of the acute phase severity scale.

Of the 165 patients studied, 139 (84%) reported symptoms at 1-year follow-up, while only 101 (61%) reported symptoms at 2-year follow-up.

**Table 1. Patients characteristics at the acute infection, vaccination status, and second infection occurrence.**

| Characteristic | N = 165[1] |
|---|---|
| Sex | |
| Female | 88 (53%) |
| Male | 77 (47%) |
| Age (years) | 53 (42, 62) |
| Body mass index (kg/m$^2$) | 24.9 (22.6, 28.0) |
| Body mass index category | |
| *Underweight* | 6 (3.6%) |
| *Normal weight* | 79 (48%) |
| *Overweight* | 55 (33%) |
| *Obese* | 25 (15%) |
| Smoking habits | |
| *Never smoked* | 98 (59%) |
| *Ex-smoker* | 41 (25%) |
| *Smoker* | 26 (16%) |
| Hypertension | 47 (28%) |
| Diabetes | 13 (7.9%) |
| Cardiovascular disease | 15 (9.1%) |
| Cerebrovascular disease | 4 (2.4%) |
| Malignant tumor | 9 (5.5%) |
| Chronic obstructive pulmonary disease | 6 (3.6%) |
| Chronic kidney disease | 1 (0.6%) |
| Acute phase severity | |
| *Scale 1: Not hospitalized, asymptomatic* | 6 (3.6%) |
| *Scale 2: Not hospitalized, symptomatic* | 63 (38%) |
| *Scale 3: Hospitalized, not requiring oxygen* | 21 (13%) |
| *Scale 4: Hospitalized, requiring oxygen (nasal cannula)* | 37 (22%) |
| *Scale 5: Hospitalized, requiring oxygen (NIV)* | 36 (22%) |
| *Scale 6: Hospitalized, ICU* | 2 (1.2%) |
| Corticosteroid | 5 (3.0%) |
| Antivirals | 100 (61%) |
| Antibiotics | 67 (41%) |
| Heparin | 49 (30%) |
| Received COVID-19 vaccine | 160 (97%) |
| Completed doses of COVID-19 vaccine | |
| *0* | 5 (3.0%) |
| *1* | 1 (0.6%) |
| *2* | 27 (16%) |
| *3* | 132 (80%) |
| Second infection | 17 (10%) |
| Second infection severity | |
| *None* | 148 (90%) |
| *Scale 1: Not hospitalized, asymptomatic* | 1 (0.6%) |
| *Scale 2: Not hospitalized, symptomatic* | 16 (9.7%) |
| *Scale 3: Hospitalized, not requiring oxygen* | 0 (0%) |
| *Scale 4: Hospitalized, requiring oxygen (nasal cannula)* | 0 (0%) |
| *Scale 5: Hospitalized, requiring oxygen (NIV)* | 0 (0%) |

(*Continued*)

**Table 1.** (Continued)

| Characteristic | N = 165[1] |
|---|---|
| *Scale 6*: *Hospitalized, ICU* | 0 (0%) |

[1]n (%); Median (IQR)

Fig 1 shows how categories of Long-COVID at 1-year follow-up and at 2-year follow-up develop based on vaccination and second infection status.

Of the patients with Long-COVID symptoms at 2-year follow-up, most patients (80, 49%) had Long-COVID at 1-year follow-up, had received the SARS-CoV-2 vaccine, and had not experienced a second infection between the 2 follow-ups.

Table 2 and Table 3 show the results from logistic generalized additive models of presenting Long-COVID at 2-years follow-up.

The analysis showed that having Long-COVID at the 1-year follow-up and contracting a second infection results in significant risk factors for presenting Long-COVID at the 2-year follow-up. This finding was consistent across most symptom subdomains, as shown in

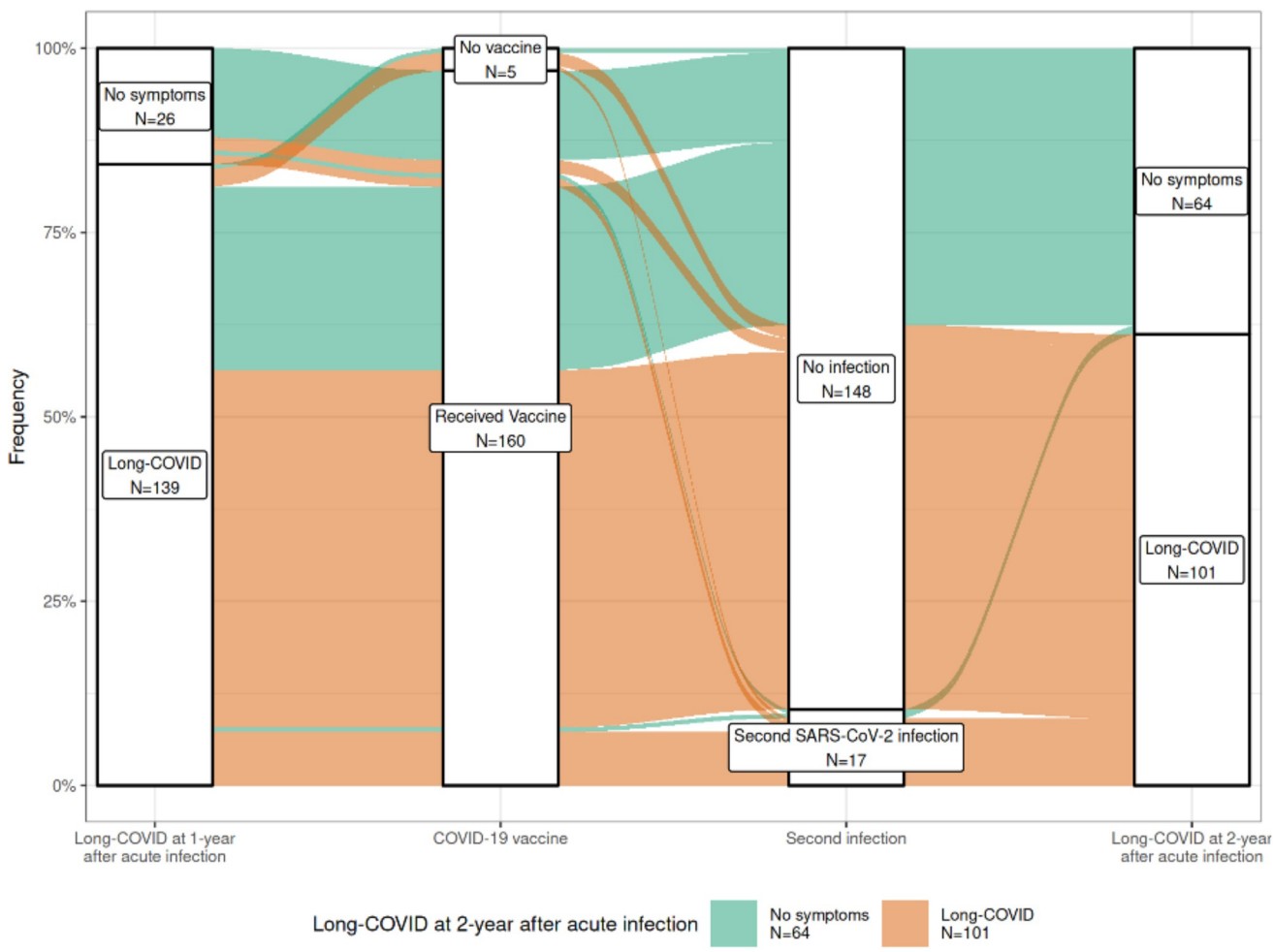

**Fig 1. Alluvial plot showing prevalence of Long-COVID at 1- and 2-year after acute infection, stratified by vaccination status and second infection occurrence.**

**Table 2. Logistic generalized additive model of any symptom of Long-COVID at 2-year after the acute infection.**

| Characteristic | log (OR)[1] | 95% CI[1] | p-value |
|---|---|---|---|
| Had symptoms 1 year after acute infection | 2.3 | 1.2, 3.5 | **<0.001** |
| Sex | | | |
| *Female* | — | — | |
| *Male* | -0.02 | -0.73, 0.70 | >0.9 |
| Acute phase severity | | | |
| *Scale 1–2: Not hospitalized* | — | — | |
| *Scale 3–6: Hospitalized* | -0.26 | -1.1, 0.55 | 0.5 |
| Second infection | 2.3 | 0.47, 4.1 | **0.013** |
| Received COVID-19 vaccine | -0.54 | -2.8, 1.8 | 0.6 |
| Age smooth[2] | | | 0.072 |

[1]OR = Odds Ratio, CI = Confidence Interval

[2]Smooth term by low rank isotropic smoothers of age in years

Table 3, a with some exceptions: sleep disorders did not show these risk factors, and the significance of a second infection varied by subdomain.

Interestingly, vaccination did not result in a protective factor, although very few patients in our sample were unvaccinated.

## Discussion

During the COVID emergency, telemedicine became essential as hospitals and community care services were overwhelmed, and COVID-19 patients were isolated at home along with the rest of the population to contain the spread of pandemic. Furthermore, telemedicine proved useful in monitoring long lasting symptoms after SARS-CoV-2 infection [8]. These long-term consequences of COVID-19, known as long-COVID, have a significant detrimental impact on patients' functioning and overall quality of life.

Our telephone surveillance served as an operative mode of telemedicine, enabling remote monitoring of the health status of patients who had experienced acute SARS-CoV2 and were at risk of developing long-COVID. More broadly, telemedicine facilitates "home health care" through a remote follow-up over time. In cases of long-COVID, telemedicine addresses multiple health needs, providing continuity and equity in access to health care [10].

To the best of our knowledge, this study stands out as one of the few that includes a 2-year follow-up on Long-COVID symptoms. We found that 84% of the patients exhibited symptoms one year after the acute infection, but this decreased to 61% after a 2-year follow-up. Although reinfections were uncommon during the follow-up period, patients who did experience a second infection had a higher risk of Long-COVID symptoms.

Evidence on Long-COVID following SARS-CoV-2 reinfections is still limited. According to our study the persistence of symptoms post COVID-19 is more frequent after the first infection than after the subsequent ones. Older patients, who often have pre-existing comorbidities and may experience more severe acute infections, have a higher cumulative risk of Long-COVID proportional to the number of infections [12].

The severity of COVID-19 is associated with the risk of developing Long-COVID. Critical disease, acute respiratory distress syndrome, long-term ventilator support and multiorgan impairment caused by COVID-19 are considered risk factors for persistent symptoms [4]. Severe COVID-19, requiring ICU admission and hospitalization, is a known risk factor described by Tsampasian et al. for Long-COVID [13]. However, Long-COVID also occurs in

**Table 3. Logistic generalized additive model of specific symptom domains of Long-COVID at 2-year after the acute infection.**

| Characteristic | Neurological and cognitive impairments | | | Fatigue and weakness | | | Movement impairments | | | Muscle and joint pain | | | Respiratory disorders | | | Sensory alterations | | | Sleep disorders | | |
|---|---|---|---|---|---|---|---|---|---|---|---|---|---|---|---|---|---|---|---|---|---|
| | Log (OR)[1] | 95% CI[1] | p-value | Log (OR)[1] | 95% CI[1] | p-value | Log (OR)[1] | 95% CI[1] | p-value | Log (OR)[1] | 95% CI[1] | p-value | Log (OR)[1] | 95% CI[1] | p-value | Log (OR)[1] | 95% CI[1] | p-value | Log (OR)[1] | 95% CI[1] | p-value |
| Had symptoms 1 year after acute infection | 1.2 | 0.43, 2.1 | **0.003** | 1.9 | 1.1, 2.8 | **<0.001** | 1.7 | 0.64, 2.8 | **0.002** | 2.0 | 1.1, 2.9 | **<0.001** | 1.7 | 0.87, 2.5 | **<0.001** | 2.7 | 1.5, 3.8 | **<0.001** | 0.69 | -0.10, 1.5 | 0.087 |
| Sex | | | | | | | | | | | | | | | | | | | | | |
| *Female* | — | — | | — | — | | — | — | | — | — | | — | — | | — | — | | — | — | |
| *Male* | 0.24 | -0.57, 1.0 | 0.6 | -0.23 | -0.97, 0.52 | 0.5 | -0.40 | -1.5, 0.66 | 0.5 | 0.32 | -0.47, 1.1 | 0.4 | -0.44 | -1.3, 0.38 | 0.3 | -0.17 | -1.1, 0.74 | 0.7 | -0.52 | -1.3, 0.26 | 0.2 |
| Acute phase severity | | | | | | | | | | | | | | | | | | | | | |
| *Scale 1-2: Not hospitalized* | — | — | | — | — | | — | — | | — | — | | — | — | | — | — | | — | — | |
| *Scale 3-6: Hospitalized* | -0.43 | -1.3, 0.43 | 0.3 | 0.25 | -0.60, 1.1 | 0.6 | 0.59 | -0.67, 1.8 | 0.4 | -0.65 | -1.5, 0.22 | 0.14 | 0.80 | -0.19, 1.8 | 0.11 | -0.38 | -1.4, 0.63 | 0.5 | 0.14 | -0.72, 0.99 | 0.8 |
| Second infection | 0.60 | -0.61, 1.8 | 0.3 | 1.2 | -0.04, 2.5 | 0.057 | 1.0 | -0.55, 2.6 | 0.2 | 1.7 | 0.53, 3.0 | **0.005** | 2.7 | 1.3, 4.2 | **<0.001** | 2.2 | 0.75, 3.6 | **0.003** | 0.18 | -1.1, 1.4 | 0.8 |
| Received COVID-19 vaccine | -1.2 | -3.3, 0.91 | 0.3 | 0.85 | -1.5, 3.2 | 0.5 | -0.85 | -3.3, 1.6 | 0.5 | 0.65 | -1.7, 3.0 | 0.6 | 43 | -58,822,389, 58,822,475 | >0.9 | -1.6 | -3.7, 0.52 | 0.14 | 0.24 | -2.0, 2.5 | 0.8 |
| Age smooth[2] | | | 0.093 | | | 0.12 | | | 0.2 | | | 0.9 | | | 0.2 | | | 0.8 | | | >0.9 |

[1]OR = Odds Ratio, CI = Confidence Interval
[2]Smooth term by low rank isotropic smoothers of age in years

non-hospitalized patients with mild acute disease [14], though less frequently than in severe acute SARS-CoV-2 infections. In our population SARS-CoV-2 infection were mild and symptomatic but hospitalization was not necessary.

The risk of Long-COVID is also influenced by SARS-CoV-2 variants [15], particularly decreasing with the spread of the Omicron variant [4]. Reinfections have become more common following the emergence of the Omicron variant, and their frequency has increased [9].

Vaccination and anti-SARS-CoV-2 immunity protect against Long-COVID, suggesting that reinfections with SARS-CoV-2 could reduce the probability of developing Long-COVID [15]. Several studies have clearly demonstrated the protective effect of vaccination against the onset of Long-COVID [12]. In our study, however, we did not find this positive association, likely due to the small number of unvaccinated patients in our sample.

Our findings highlight no gender difference in Long-COVID onset, although according to several studies the female sex is more frequently affected by persistent symptoms post-acute infection [16, 17]. According to the results of this study, COVID-19 seems to be more frequent in people with chronic comorbidities such as hypertension, diabetes and cardiovascular diseases, as suggested in scientific literature [18]. Lifestyle factors like smoking have a higher risk of long lasting symptoms: current smokers have a higher risk compared to those who have never smoked, likely due [16] to compromised immune and cardiovascular systems [19]. Our study could not confirm this association because most patients were non-smokers. Similarly, obesity and overweight were not associated with a higher disease risk in our cohort.

More than 50% of patients reported symptoms of Long-COVID at 2-year follow-up. Long-COVID can present heterogeneously, with respiratory symptoms, muscular and joint complains, neurological and sleep disorders and gastrointestinal symptoms, significantly impacting daily life. These findings highlight the importance of a long-term follow-up for post-acute COVID-19 patients to ensure multidisciplinary patient care, or a specialized referral in case of a specific disorder. A period of rehabilitation may also be necessary [20]. Remote surveillance is an effective, simple, and well-accepted telemedicine follow-up method, improving the quality of life for patients with previous COVID19.

The sample size of our study is small, necessitating further research on a wider cohort with longer follow-ups. While telemedicine has some barriers, such as accessibility, maintenance costs and insufficient legal regulations, these obstacles are surmountable given its usefulness in monitoring many clinical conditions.

## Conclusion

Through the COD19 platform, we monitored patients post-COVID-19 for two years, identifying the persistence of Long-COVID symptoms. The potential value of telemedicine platforms is evident in pandemic situations and other conditions where isolation is essential. Telemedicine has proven to be an innovative tool for effectively monitoring patients over an extended period and for early diagnosis of conditions requiring treatment. It represents a future challenge in healthcare

## Acknowledgments

The authors thank Dr. Nicola Bladen for English revision of the manuscript.

## Author Contributions

**Conceptualization:** Andrea Foppiani, Valeria Calcaterra, Gianvincenzo Zuccotti.

**Formal analysis:** Andrea Foppiani.

**Investigation:** Andrea Foppiani, Chiara Montanari, Sara Zanelli, Michele Davide Maria Lombardo.

**Methodology:** Andrea Foppiani, Chiara Montanari, Sara Zanelli, Michele Davide Maria Lombardo, Valeria Calcaterra, Gianvincenzo Zuccotti.

**Supervision:** Andrea Foppiani, Valeria Calcaterra, Gianvincenzo Zuccotti.

**Writing – original draft:** Andrea Foppiani, Chiara Montanari, Sara Zanelli, Michele Davide Maria Lombardo, Valeria Calcaterra, Gianvincenzo Zuccotti.

**Writing – review & editing:** Andrea Foppiani, Chiara Montanari, Valeria Calcaterra, Gianvincenzo Zuccotti.

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
