## [Decision Letter · Decision Letter 0]

10 Jun 2024

PONE-D-24-17457Monitoring Long-COVID Symptoms at a Two-Year Follow-Up Through a Telemedicine PlatformPLOS ONE

Dear Dr. CALCATERRA,

Thank you for submitting your manuscript to PLOS ONE. After careful consideration, we feel that it has merit but does not fully meet PLOS ONE’s publication criteria as it currently stands. Therefore, we invite you to submit a revised version of the manuscript that addresses the points raised during the review process.

The reviewers suggest minor revision to the manuscript. Furthermore, the manuscript will need English language editing to improve readability.

We look forward to receiving your revised manuscript.

Kind regards,

Seth Kwabena Amponsah, PhD

Academic Editor

PLOS ONE

Journal Requirements:

Reviewers' comments:

Reviewer's Responses to Questions

**Comments to the Author**

1. Is the manuscript technically sound, and do the data support the conclusions?

Reviewer #1: Yes

Reviewer #2: Yes

2. Has the statistical analysis been performed appropriately and rigorously? 

Reviewer #1: Yes

Reviewer #2: Yes

3. Have the authors made all data underlying the findings in their manuscript fully available?

Reviewer #1: Yes

Reviewer #2: Yes

4. Is the manuscript presented in an intelligible fashion and written in standard English?

Reviewer #1: Yes

Reviewer #2: No

5. Review Comments to the Author

Reviewer #1: An interesting read and very easy to follow. I will recommend to readers.

However, the manuscript will benefit from an independent English editor. Few sentences here and there will require rephrasing and a Grammarly check.

Reviewer #2: I have recommended rephrasing the title and the short title. In addition, the manuscript need an English language editing since there are a lot of grammatical errors which affect reading.

Kindly see the attached document for the changes.

6. PLOS authors have the option to publish the peer review history of their article (what does this mean?). If published, this will include your full peer review and any attached files.

Reviewer #1: No

Reviewer #2: No

---

## [Author Response · Author response to Decision Letter 0]

13 Jun 2024

Dear Editor,

with pleasure, we resubmit the revised version of our manuscript entitled “Long-COVID Symptom Monitoring: Insights from a Two-Year Telemedicine Study” which I and co-authors submit for to be taken into consideration for PLOSONE.

A point-by-point response to the Reviewer’s comments is provided below (the changes are tracked in blue).

As suggested by reviewer, the title was modified and the English language revised. 

Neither the manuscript nor any significant part of it has been previously published or is under consideration for publication elsewhere.

We wish to thank the reviewers for their appreciation and we hope that the manuscript will be suitable for publication in the present form.

Thank you for your kind attention and consideration.

Milano, 12-06-24

Sincerly yours

Valeria Calcaterra

Reviewers' comments:

Reviewer's Responses to Questions

Comments to the Author

1. Is the manuscript technically sound, and do the data support the conclusions?

Reviewer #1: Yes R: thank you for your positive comments

Reviewer #2: Yes R: thank you for your positive comments

2. Has the statistical analysis been performed appropriately and rigorously? 

Reviewer #1: Yes R: thank you for your positive comments

Reviewer #2: Yes R: thank you for your positive comments

3. Have the authors made all data underlying the findings in their manuscript fully available?

Reviewer #1: Yes R: thank you for your positive comments

Reviewer #2: Yes R: thank you for your positive comments

4. Is the manuscript presented in an intelligible fashion and written in standard English?

Reviewer #1: Yes R: thank you for your positive comments

Reviewer #2: No R: as suggested the manuscript has been revised by a native speaker (the changes are tracked in blue)

5. Review Comments to the Author

Reviewer #1: An interesting read and very easy to follow. I will recommend to readers.

However, the manuscript will benefit from an independent English editor. Few sentences here and there will require rephrasing and a Grammarly check. 

R: as suggested the manuscript has been revised by a native speaker (the changes are tracked in blue)

Reviewer #2: I have recommended rephrasing the title and the short title. In addition, the manuscript need an English language editing since there are a lot of grammatical errors which affect reading.

R: as suggested, we rephrased the title and the manuscript has been revised by a native speaker (the changes are tracked in blue)

---

## [Decision Letter · Decision Letter 1]

12 Jul 2024

Long-COVID Symptom Monitoring: Insights from a Two-Year Telemedicine Study

PONE-D-24-17457R1

Dear Dr. CALCATERRA,

We’re pleased to inform you that your manuscript has been judged scientifically suitable for publication and will be formally accepted for publication once it meets all outstanding technical requirements.

Kind regards,

Seth Kwabena Amponsah, PhD

Academic Editor

PLOS ONE

Additional Editor Comments (optional):

Reviewers' comments:

Reviewer's Responses to Questions

**Comments to the Author**

1. If the authors have adequately addressed your comments raised in a previous round of review and you feel that this manuscript is now acceptable for publication, you may indicate that here to bypass the “Comments to the Author” section, enter your conflict of interest statement in the “Confidential to Editor” section, and submit your "Accept" recommendation.

Reviewer #1: All comments have been addressed

2. Is the manuscript technically sound, and do the data support the conclusions?

Reviewer #1: Yes

3. Has the statistical analysis been performed appropriately and rigorously? 

Reviewer #1: I Don't Know

4. Have the authors made all data underlying the findings in their manuscript fully available?

Reviewer #1: Yes

5. Is the manuscript presented in an intelligible fashion and written in standard English?

Reviewer #1: Yes

6. Review Comments to the Author

Reviewer #1: The manuscript reads better. Comments have been addressed. The reviewer commends the authors for their effort in improving the manuscript

7. PLOS authors have the option to publish the peer review history of their article (what does this mean?). If published, this will include your full peer review and any attached files.

Reviewer #1: No

---

## [Editor Report · Acceptance letter]

17 Jul 2024

PONE-D-24-17457R1 

PLOS ONE

Dear Dr. Calcaterra, 

I'm pleased to inform you that your manuscript has been deemed suitable for publication in PLOS ONE. Congratulations! Your manuscript is now being handed over to our production team.

Kind regards, 

on behalf of

Prof. Seth Kwabena Amponsah 

Academic Editor

PLOS ONE